# Evaluation of the Performance of CLM5.0 in Soil Hydrothermal Dynamics in Permafrost Regions on the Qinghai–Tibet Plateau

Shuhua Yang [1,2], Ren Li [2,*], Lin Zhao [2,3], Tonghua Wu [2,4], Xiaodong Wu [2], Yuxin Zhang [3], Jianzong Shi [2] and Yongping Qiao [2]

1   School of Environmental Sciences, Nanjing Xiaozhuang University, Nanjing 211171, China
2   Cryosphere Research Site on the Qinghai-Tibet Plateau, State Key Laboratory of Cryospheric Science, Northwest Institute of Eco-Environment and Resource, Chinese Academy of Sciences, Lanzhou 730000, China
3   School of Geographical Sciences, Nanjing University of Information Science & Technology, Nanjing 210000, China
4   Southern Marine Science and Engineering Guangdong Laboratory, Guangzhou 511458, China
*   Correspondence: liren@lzb.ac.cn

**Abstract:** Soil hydrothermal dynamics are crucial processes for understanding the internal physical conditions of the active layer in permafrost regions. It is very difficult to obtain data in permafrost regions, especially on the Qinghai–Tibet Plateau (QTP). Land surface modes (LSMs) provide an effective tool for soil hydrothermal dynamics. However, it is necessary to evaluate the simulation performance before using them. Here, we used two in situ sites along with the latest version of the Community Land Model (CLM5.0) to evaluate the simulated performance in the soil hydrothermal parameters of the model in permafrost regions on the QTP. Meanwhile, the effects of soil properties, thermal roughness length, and the freeze–thaw process on the simulation results were investigated. The results showed that CLM5.0 can capture the dynamic changes in soil hydrothermal changes well in permafrost regions on the QTP. Soil moisture and thermal conductivity were more sensitive to soil properties and the freeze–thaw process, while the thermal roughness length had a greater effect on soil temperature. Notably, although we improved the soil properties and thermal roughness length, there were still some errors, especially in the soil moisture and soil thermal conductivity. It may be caused by inappropriate hydrothermal parameterizations of the model, especially the soil thermal conductivity, hydraulic conductivity, unfrozen water scheme, and snow schemes. There is an urgent need for collaboration between experts in permafrost science, hydrological science, and modelers to develop the appropriate schemes for permafrost regions and enhance the LSMs.

**Keywords:** soil hydrothermal; permafrost; CLM; Qinghai–Tibet Plateau

## 1. Introduction

The Qinghai–Tibet Plateau (QTP) is known as the "Third Pole of the Earth", with an average elevation of approximately 4000 m [1], and the "Water Tower of Asia" [2]. Due to its unique climatic and geomorphological conditions, there are widespread cryosphere components, such as glaciers, permafrost, and snow [2,3]. Permafrost covered areas of $1.06 \times 10^6$ km$^2$ (approximately 40% of the total land area) on the QTP [4]. It is significantly sensitive to climate warming, and changes in permafrost can notably affect ecology, hydrology, engineering, and construction [3,5]. In recent decades, with climate warming, some changes have occurred in permafrost regions on the QTP, such as the active layer thickness [6,7], ground temperature of permafrost [6], permafrost extent [4], and thermokarst lakes [8]. These changes can cause surface settlement and serious hazards to the stability of the Qinghai–Tibet Railway and Qinghai–Tibet Highway [9,10]. Additionally, previous studies have shown that the amount of organic carbon stored in permafrost is approximately twice that of the atmospheric carbon content [11]. The degradation of permafrost causes organic carbon to be decomposed into carbon dioxide and methane and released into the

atmosphere, which creates positive feedback on the current global warming [12]. Therefore, it is very useful to investigate the physical conditions of permafrost for ecology, hydrology, engineering, and construction on the QTP. The active layer is located in the upper layer of the permafrost, which melts in summer and freezes in winter. It is the direct contact surface of the energy–water exchange between permafrost and the atmosphere [13]. Therefore, the investigation of soil hydrothermal regimes within the active layer is a prerequisite for understanding permafrost changes.

Soil temperature, soil moisture, and soil thermal conductivity are three key hydrothermal properties in understanding soil physical conditions in permafrost regions. Soil temperature influences physical, ecological, and microbial processes in the soil by regulating water vapor transport and phase change [14,15]. Soil moisture affects the energy balance and climate change by changing surface albedo, and sensible and latent heat fluxes [16]. Soil thermal conductivity is one of the main factors that determine the heat transfer capacity of soil [17,18], and it is also an important input parameter for numerical simulation [7]. In situ monitoring is the most direct and efficient method to obtain these data in permafrost regions [19]. Limited by the harsh climate and geological conditions, the in situ measurements were forced on single points or small local areas [19]. To compensate for these shortcomings, in recent years, remote sensing and numerical simulation techniques have been rapidly developed and used. Remote sensing can provide large-scale and real-time data. Hence, researchers usually use Moderate-Resolution Imaging Spectroradiometer (MODIS) products to investigate surface temperature and permafrost extent characteristics [4,20]. Microwave remote sensing is widely used in soil moisture. Recently, Xing et al. (2021) used in situ measurements to assess the performance of seven satellite soil moisture products and demonstrated that surface soil moisture of the European Space Agency Climate Change Initiative (ESA CCI) has a superior performance in permafrost regions on the QTP [21]. Additionally, some researchers have employed InSAR data to study surface subsidence [9]. However, remote sensing can only acquire topsoil data (i.e., 5–10 cm soil depth), which limits the understanding of deep soil conditions in permafrost regions. In contrast, numerical simulations can not only generate soil hydrothermal data at different depths at the regional scale by combining observations and remote sensing but can also provide a useful tool for predicting future changes [22,23].

As one of the important tools of numerical simulation, the land surface models (LSMs) integrate the physical processes of soil, ecology, and hydrology, and have been widely used over the past several decades, such as CLM (the community land model), Noah–MP (Noah with multi-parameterization options), SiB2 (simple biosphere model), and JULES (Joint UK Land Environment Simulator) [22–24]. However, the LSMs were not originally developed for permafrost. With the growing understanding of the important role of frozen ground, some parameterizations for frozen ground have been subsequently incorporated into LSMs, such as ice–water phase change [25], frozen fronts [26], and thermal conductivity schemes during the freezing period [27]. However, the performance of different LSMs differs between regions due to the differences in the parameterizations used in the models [28]. Most of the LSMs have good simulation performance in soil temperature, whereas the simulation error in soil moisture is larger [29,30]. Moreover, the soil temperature in the cold season was significantly underestimated in mid–low-latitude and high-elevation regions, especially the QTP [15,30]. Previous studies have demonstrated that these errors may be related to soil organic carbon, soil stratification structure, and the parameterization schemes within the model [30–32]. Studies have suggested that different soil thermal conductivity schemes can lead to errors of 1–3 °C in soil temperature [33]. In particular, the incorporation of organic matter into the LSM significantly improved the simulation accuracy of soil moisture [27,34]. Furthermore, previous studies have shown that surface roughness is one of the main factors resulting in heat flux transfer into the ground [33]. An improved thermal roughness length can reduce errors in soil thermal properties [35].

The CLM model is one of the most comprehensive LSMs, which integrates the soil, hydrological, and ecological sections. The latest version of CLM, CLM5.0, incorporates

many new and updated processes and parameterizations based on CLM4.5, including snow density, plant hydraulics, hydraulic redistribution, river model, and carbon and nitrogen cycling [22]. In addition, an outstanding improvement is that CLM5.0 increases the number of soil layers at a depth of 3 m, which facilitates the study of permafrost. Recent studies have indicated that comparing the performance between CLM5.0 and CLM4.5 in seasonally frozen ground regions on the QTP indicated that CLM5.0 has better performances in soil temperature and some key parameters than CLM4.5 [31,36]. However, the simulation performance of CLM5.0 in soil hydrothermal dynamics in permafrost regions on the QTP is not conducted. In this paper, we evaluate the performance of the CLM5.0 based on two in situ sites in permafrost regions on the QTP. Additionally, we also conducted four sensitivity experiments to discuss the effects of the soil properties, thermal roughness length, and the freeze–thaw process on the hydrothermal simulation results. In Section 2, we introduce the measurement data, CLM5.0, experimental design, and statistical methods. In Section 3, we evaluate the simulation performance of CLM5.0 and the sensitivity of soil hydrothermal parameters to soil properties, thermal roughness length, and the freeze–thaw process. A discussion of the possible factors and uncertainty of these results is provided in Section 4. Finally, the conclusions of the study are summarized in Section 5.

## 2. Materials and Methods

### 2.1. In Situ Sites and Measurements

In this study, the Tanggula and Beiluhe sites were selected to evaluate the simulation performance in CLM5.0 in permafrost regions on the QTP. The two sites were located in a continuous permafrost region in the central part of the QTP (Figure 1). The Tanggula site (33.07°N, 91.93°E) is currently the highest altitude in situ monitoring site in permafrost regions on the QTP, with an altitude of 5100 m. The main vegetable type is the alpine meadow, the vegetation coverage is approximately 51%, and the average annual temperature and total precipitation at the Tanggula site were about −4.4 °C and 375 mm, respectively [30]. The Beiluhe site (34.82°N, 92.92°E) is located upstream of Beiluhe Basin over the center of the QTP. Alpine swamp is the main vegetable type and vegetable coverage, with 82% vegetation coverage. Its average annual temperature and total precipitation were about −3.0 °C and 415 mm, respectively [30].

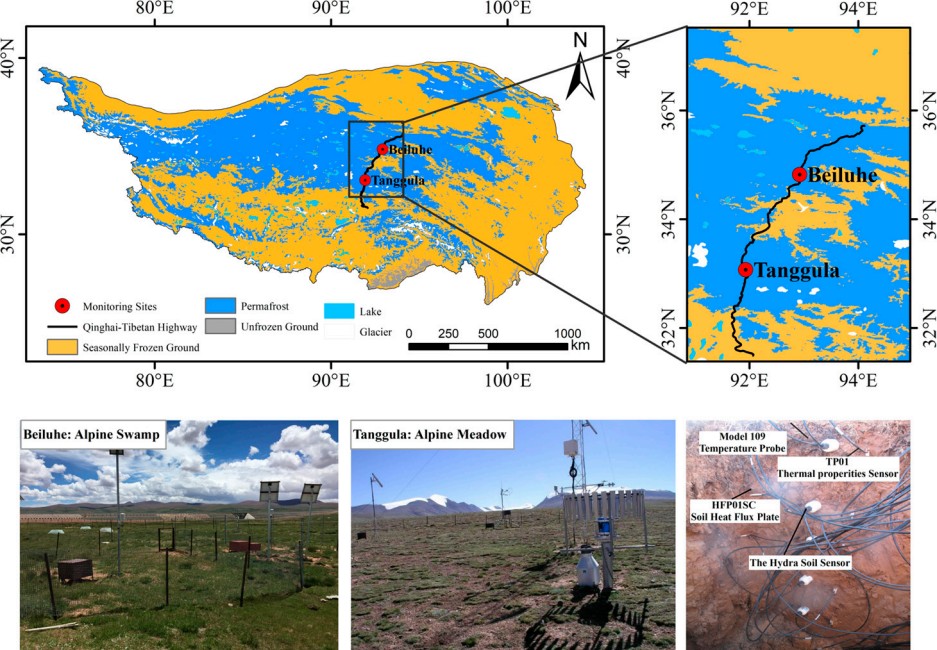

**Figure 1.** Distribution of the study sites and the surrounding vegetation conditions (the frozen ground map is derived from [4]).

The atmospheric forcing data at the Beiluhe site were derived from Li et al. (2020) [30], which covered the period from 20 August 2009 to 19 August 2010. The available observed data with high quality at the Tanggula site were collected from 20 August 2006 to 19 August 2007 (Table 1). Soil temperature, soil moisture, and soil thermal conductivity were used to evaluate the performance of the simulations by CLM5.0. Soil temperatures at different depths were monitored by 109 and 105 T temperature probes (Campbell Scientific, Inc., Logan, UT, USA) with an accuracy of 0.1 °C. The Stevens Hydro Soil Sensor (Stevens Water Monitoring System, Inc., Portland, OR, USA) with an accuracy of ±3% was employed to monitor soil moisture. Probes were installed at 10–20 cm intervals in shallow layers and 20–80 cm intervals with increasing depth [32]. Soil thermal conductivity was ascertained through the soil heat flux and temperature gradient between 5 and 10 cm soil depths. The detailed theory and calculation method were introduced in the articles [37].

**Table 1.** Information on the two sites in permafrost regions on the QTP.

| Site Name | Longitude /°E | Latitude /°N | Altitude /m | Vegetation Type | Coverage /% | Temporal Coverage | Soil Temperature Depth/cm | Soil Moisture Depth/cm |
|-----------|---------------|--------------|-------------|-----------------|-------------|-------------------|---------------------------|------------------------|
| Beiluhe * | 92.92 | 34.82 | 4656 | Alpine swamp | 82 | 20 August 2009–20 August 2010 | 5, 30, 50, 60, 90, 120, 150, 180, 220 | 5, 10, 20, 50 |
| Tanggula | 91.93 | 33.07 | 5100 | Alpine meadow | 51 | 20 August 2005–20 August 2007 | 5, 10, 20, 50, 70, 90, 105, 140, 175, 210 | 5, 10, 20, 35 |

* The atmospheric forcing data at the Beiluhe site are derived from [30].

### 2.2. Methods

### 2.2.1. The Description of CLM5.0

The CLM is released by the National Center for Atmosphere Research (NCAR) and is used for land surface process simulation on the regional or single–point scale. CLM5.0 is the latest version of the CLM and is an upgraded version of CLM4.5 [22]. Compared with CLM4.5, many processes and parameterizations have been updated and introduced in CLM5.0, including snow density, hydraulic redistribution, the river model, and nitrogen cycling. Specifically, the number of soil stratifications at a depth of 3 m below the soil is added in CLM5.0, which is helpful to the simulation of the active layer. CLM5.0 includes 25 layers of soil vertical discretization, of which 20 layers are hydrologically and biogeochemically active [38]. For more detailed technical descriptions, see Lawrence et al. (2019) [22].

In CLM5.0, the traditional hydrothermal equation is adopted. That is, the one-dimensional heat transfer equation is used for soil heat transfer, while the modified Richard's equation is used for soil moisture transfer [39]. In the model solution, the temperature profile is calculated first without phase change and then readjusted to the phase change [25]. Heat advection associated with water infiltrating into the soil is not considered [38]. The soil temperatures are evaluated to determine if phase change will take place, as follows:

$$T > T_f \text{ and } w_{ice} > 0, \text{ thawing,} \tag{1}$$

$$T < T_f \text{ and } w_{liq} > w_{liq, max}, \text{ freezing,} \tag{2}$$

where $T$ is the soil temperature (K), $T_f$ is the freezing temperature (273.15 K), $w_{ice}$ and $w_{liq}$ are the masses of ice and liquid water (kg·m$^{-2}$), respectively, and $w_{liq, max}$ is the maximum liquid water when the soil temperature is below the freezing temperature.

Unlike other soils, liquid water still exists at negative temperatures (i.e., unfrozen water) in permafrost. To truly capture this phenomenon, the maximum liquid water was proposed in [25], and the equation is as follows:

$$w_{liq,\,max,i} = \triangle z_i \theta_{sat,i} \left[ \frac{10^3 L_f \left( T_f - T_i \right)}{g T_i \psi_{sat,i}} \right]^{-1/B_i}, \; T_i < T_f, \tag{3}$$

where $\triangle z_i$ is the thickness of the soil layer $i$ (mm), $\theta_{sat,i}$ is the saturated water content (mm$^3$·mm$^{-3}$), $L_f$ is the latent heat of fusion (J·kg$^{-1}$), $\psi_{sat,i}$ is matric potential (mm), and $g$ and $B_i$ are the gravitational acceleration (m·s$^{-2}$) and Clapp and Hornberger exponents, respectively.

The soil thermal conductivity, $\lambda$ (W·m$^{-1}$·K$^{-1}$), is an improved scheme by incorporating organic matter into Farouki's scheme [27], as follows:

$$\lambda = \begin{cases} K_e \left( \lambda_{sat} - \lambda_{dry} \right) + \lambda_{dry}, S_r > 10^{-7} \\ \lambda_{dry}, \; S_r \leq 10^{-7} \end{cases}, \tag{4}$$

$$K_e = \begin{cases} \log S_r + 1, \; T \geq T_f \\ S_r, \; T < T_f \end{cases}, \tag{5}$$

$$\lambda_{sat} = \lambda_w^{\frac{\theta_l \cdot \theta_{sat}}{\theta_l + \theta_{ice}}} \lambda_s^{1 - \theta_{sat}} \lambda_{ice}^{\theta_{sat} \left( 1 - \frac{\theta_l}{\theta_l + \theta_{ice}} \right)}, \tag{6}$$

$$\lambda_s = (1 - f_{om}) \lambda_{s,min} + f_{om} \lambda_{s,om}, \tag{7}$$

$$\lambda_{dry} = (1 - f_{om}) \lambda_{dry,min} + f_{om} \lambda_{dry,om} \tag{8}$$

$$\lambda_{dry,min} = \frac{0.135 \rho_d + 64.7}{2700 - 0.947 \rho_d} \tag{9}$$

where $\lambda_{sat}$ is the saturated thermal conductivity (W·m$^{-1}$·K$^{-1}$), $K_e$ is the Kersten number, $S_r$ is the wetness of the soil with respect to saturation, $\lambda_s$ is the thermal conductivity of soil solids (W·m$^{-1}$·K$^{-1}$), $\lambda_{s,min}$ is the mineral soil solid thermal conductivity (W·m$^{-1}$·K$^{-1}$), $\lambda_{dry}$ is the dry thermal conductivity (W·m$^{-1}$·K$^{-1}$), $\lambda_{dry,min}$ is the thermal conductivity of dry soil (W·m$^{-1}$·K$^{-1}$), $\lambda_w$ (0.57W·m$^{-1}$·K$^{-1}$) and $\lambda_{ice}$ (2.29 W·m$^{-1}$·K$^{-1}$) are the thermal conductivities of water and ice, respectively, $f_{om}$ is the soil layer organic matter fraction, the bulk density $\rho_d = 2700 \, (1 - \theta_{sat,i})$, and %$_{sand}$, %$_{clay}$, $\theta_l$, and $\theta_{ice}$ are the sand, clay, and volumetric liquid water and ice contents, respectively.

The detailed technical description of CLM5.0 is available online (http://www.cesm. ucar.edu/models/cesm2/land/, 1 June 2022). Overall, CLM5.0 has completed and well-developed hydrothermal processes and can be used as an effective tool to study soil hydrothermal processes.

### 2.2.2. Model Setup

The CLM is a complex system that includes biogeophysical and biochemical processes. Carbon and nitrogen cycles in biogeochemical processes take more than 1000 years for spin-up to reach equilibrium, which requires a high computational performance [40]. Biogeochemical processes are not involved in this study. To improve the computational efficiency, we selected the Satellite Phenology Model (CLMSP) for single-point offline experiments. A 30-year spin-up was conducted to reach the equilibrium of the hydrothermal regime, and the equilibrium values were used as the initial condition for the final simulations in CLM5.0.

Note that the first-year data were spun-up at the Tanggula site, and one-year data to recycle at the Beiluhe site due to the forcing data being only one year. The soil column is divided into multiple layers, vertical and integrated downward over each layer with an upper boundary condition of the infiltration flux into the topsoil layer and a zero-flux lower boundary condition at the bottom of the soil column [41]. In CLM5.0, soil thickness

can vary within a range of 0.4 to 8.5 m depth. The default model soil layer resolution is increased, especially within the top 3 m, which is very helpful for this study of the active layer in permafrost regions [38]. Since the depth of in situ measurements is 210 cm and the default soil stratification structure is close to observations, therefore, the default soil stratification structure is used in this study. In addition, as the main objective of this study is to evaluate the performance of the CLM5.0 in permafrost regions on the QTP, we use the default parameters and archived data for the others.

### 2.2.3. Experimental Designs

Previous studies have shown that soil properties determine soil thermal and hydrologic properties, including soil texture and organic matter [27,42]. In CLM5.0, soil texture data are derived from the International Geosphere–Biosphere Programme (IGBP) soil dataset, which is based on 4931 soil mapping units and includes sand and clay content for each soil layer [43]. Soil organic matter data were merged from the Global Soil Profile data (ISRIC–WISE) (http://www.isric.org, 1 June 2022) and the Northern Circumpolar Soil Carbon Database (https://bolin.su.se/data/ncscd/, 1 June 2022). However, these data lacked soil samples from the QTP, especially in permafrost regions, so there was a large error in soil properties. In this study, to investigate the effect of soil properties on soil hydrothermal simulated results, we replaced the default soil properties with observations at two sites, including sand, clay, and soil organic matter density contents. Table 2 presents information on the soil properties at the Tanggula and Beiluhe sites. Note that soil property data in the surface file are only provided up to the 10th layer (1.36 m), and the rest of the data are derived from the 10th layer [38].

**Table 2.** Soil property information at the Tanggula and Beiluhe sites.

| Depths (cm) | Tanggula Site | | | | | | Beiluhe Site | | | | | |
| --- | --- | --- | --- | --- | --- | --- | --- | --- | --- | --- | --- | --- |
| | Sand (%) | | Clay (%) | | Organic (kg·m$^{-3}$) | | Sand (%) | | Clay (%) | | Organic (kg·m$^{-3}$) | |
| | Default | OBS | Default | OBS | Default | OBS | Default | OBS | Default | OBS | Default | OBS |
| 1 | 60 | 85.0 | 19 | 5.0 | 24.8 | 14.4 | 60 | 40.2 | 19 | 5.8 | 65.2 | 38.8 |
| 4 | 60 | 85.0 | 19 | 5.0 | 16.9 | 14.4 | 60 | 40.2 | 19 | 5.8 | 59.4 | 38.8 |
| 9 | 60 | 75.0 | 19 | 7.0 | 10.9 | 14.4 | 60 | 40.2 | 19 | 5.8 | 46.7 | 38.8 |
| 16 | 60 | 70.0 | 19 | 12.0 | 7.0 | 19.3 | 60 | 44.2 | 19 | 5.7 | 43.4 | 20.0 |
| 26 | 59 | 65.0 | 20 | 13.0 | 4.5 | 16.0 | 59 | 44.2 | 20 | 5.7 | 36.4 | 22.5 |
| 40 | 58 | 85.0 | 21 | 5.0 | 2.9 | 12.6 | 58 | 44.2 | 21 | 5.7 | 29.3 | 25.2 |
| 58 | 58 | 85.0 | 21 | 5.0 | 1.9 | 3.3 | 58 | 73.1 | 21 | 7.5 | 23.2 | – |
| 80 | 58 | 85.0 | 20 | 5.0 | 1.2 | 2.9 | 58 | 73.1 | 20 | 7.5 | 18.2 | – |
| 105 | 61 | 95.0 | 18 | 2.0 | 0.0 | 2.9 | 61 | 73.1 | 18 | 7.5 | 0.0 | – |
| >136 | 53 | 95.0 | 24 | 2.0 | 0.0 | 2.9 | 53 | 73.1 | 24 | 7.5 | 0.0 | – |

Note: "–" represents missing data.

Surface roughness significantly affects the amount of heat entering the soil layer [33]. Thermal roughness length is an essential parameter of surface roughness. A thermal roughness length scheme ($z_{oh}$) was improved by Yang et al., (2008) (Y08) [35]. Subsequently, this scheme has been widely used on the QTP [32,44]. Therefore, we also incorporated this scheme into our CLM5.0 (EXP2 in Table 3). The detailed equation is expressed by the following equations:

$$z_{oh} = \left(\frac{70v}{\mu_*}\right) \times \exp\left(-\beta\mu_*^{0.5}|T_*|^{0.25}\right), \tag{10}$$

$$v = v_0 \left(\frac{p_0}{p}\right)\left(\frac{T}{T_0}\right)^{1.754}, \tag{11}$$

$$T_* = -\frac{H}{\rho_a C_P \mu_*}, \tag{12}$$

where $v$ is the air kinematic viscosity (m$^2$·s$^{-1}$), $\mu_*$ is the friction velocity (m·s$^{-1}$), $\beta = 7.2$ m$^{-\frac{1}{2}}$·s$^{\frac{1}{2}}$·K$^{-\frac{1}{4}}$, $v_0 = 1.328 \times 10^{-5}$ m$^2$·s$^{-1}$, $p$ is the surface pressure (Pa),

$p_0 = 1.013 \times 10^5$ Pa, $T_*$ is the temperature scale (K), $T$ is the surface air temperature (K), $H$ is the sensible heat flux $(\text{W·m}^{-2})$, $\rho_a$ is the air density $(\text{kg·m}^{-3})$, and $C_P = 1004 \,\text{J·kg}^{-1}\text{·K}^{-1}$.

**Table 3.** Designs of four sensitivity experiments by CLM5.0.

| Experiments | Soil Properties | Thermal Roughness Length | Freeze–Thaw Process |
|---|---|---|---|
| CTL | Default | Default | Default |
| EXP1 | Observation | Default | Default |
| EXP2 | Observation | Y08 | Default |
| EXP3 | Observation | Y08 | Y18 |

Additionally, there are repeated freeze–thaw processes in the active layer, which influence the energy and water exchange between the soil and the atmosphere [45]. Compared to other models, although the phase change process (Equations (1)–(3)) has been incorporated into CLM 5.0 to improve the accuracy of the simulation in the frozen ground region, there are still large uncertainties [45,46]. Recently, Yang et al., (2018) (Y18) improved the freeze–thaw process by incorporating virtual temperature and phase change efficiency into CLM4.5 [46]. The virtual soil temperature ($T_v$) is expressed by the following equation:

$$T_v = \frac{10^3 L_f T_f}{10^3 L_f - g\psi_{sat}\left(\frac{\theta_l}{\theta_{sat}}\right)^{-B}},$$ (13)

The phase change efficiency ($\varepsilon$) is subsequently incorporated:

$$\varepsilon = \begin{cases} \frac{\theta_l}{\theta_{sat}}, & Freezing \\ \frac{\theta_{ice}}{\theta_{sat}}, & Thawing \end{cases},$$ (14)

It is assumed that more energy will be employed for freezing when the soil liquid water content is comparatively large, and a similar idea holds for the thawing of soil ice [46].

In the phase change process, the hypothetical ice mass ($H_m$) is calculated as:

$$H_m = \frac{c\Delta z}{L_f}(T_v - T)\varepsilon,$$ (15)

where $c$ is the volumetric heat capacity of the soil ($\text{J·m}^3\text{·K}$), and the other variables are the same as those used in the equations above. The detailed process of the phase change can be found in the CLM5.0 technical manual (http://www.cesm.ucar.edu/models/cesm2/land/, 1 June 2022).

With the above improvements, it was suggested that the simulated soil temperature and moisture by the improved model were closer to observations in seasonally frozen ground regions on the QTP [46]. Hence, we incorporated these improvements into CLM5.0 to investigate its performance in simulating soil hydrothermal parameters in permafrost regions (EXP3 in Table 3).

2.2.4. Statistical Metrics

In this study, three statistical metrics, the correlation coefficient ($R$), mean bias error ($MBE$), and root mean square error ($RMSE$), were employed to evaluate the simulation performance of CLM5.0 in permafrost regions on the QTP, as follows:

$$R = \frac{\sum_{i=1}^{N}\left(M_i - \overline{M}\right)\left(O_i - \overline{O}\right)}{\sqrt{\sum_{i=1}^{N}\left(M_i - \overline{M}\right)^2} \cdot \sqrt{\sum_{i=1}^{N}\left(O_i - \overline{O}\right)^2}}$$ (16)

$$MBE = \frac{1}{N}\sum_{i=1}^{N}(M_i - O_i)$$ (17)

$$RMSE = \sqrt{\frac{1}{N}\left(\sum_{i=1}^{N}(M_i - O_i)^2\right)} \tag{18}$$

where $M_i$ ($i = 1, 2, \ldots. N$) is the simulated value, $O_i$ is the observed value, and $\overline{M}$ and $\overline{O}$ represent the average values of simulated and observed values, respectively.

## 3. Results

### 3.1. Soil Temperature

The dynamic change in soil temperature at different depths derived from the simulation by CLM5.0 against the observations at the Tanggula site is shown in Figure 2. The related statistics of the soil temperature at the two sites are listed in Table 4. CLM5.0 can capture the temporal variation characteristics of the soil temperature at the Tanggula site well, with an R-value of approximately 0.9, especially at the shallow layer (0–40 cm) with an R-value reaching 0.99 (Table 4). It is worth noting that CLM5.0 can not only simulate the general trend of soil temperature, but can also capture some jumps very well (e.g., on 31 December 2006 in Figure 2). However, the soil temperature is systematically underestimated during the freezing period (MBE < 0 °C) and overestimated during the thawing period (MBE > 0 °C), and the errors increase with soil depth (Figure 2 and Table 4).

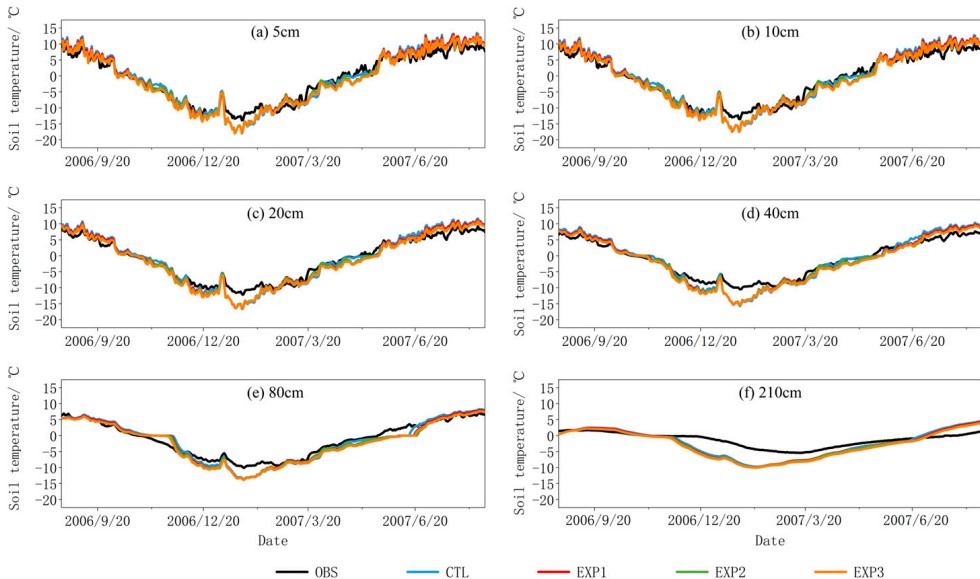

**Figure 2.** Soil temperature changes between observed and simulated performance at different depths at the Tanggula site.

Compared to the default soil property data, the simulation accuracy of soil temperature can be improved by using the measured soil property data (EXP1) with a RMSE decrease by about 0.2 °C on average at the 5–40 cm depth (Table 4). However, the EXP1 experiment did not perform well in the deeper soils (>40 cm). To further improve the simulation accuracy in soil temperature, we incorporated the thermal roughness length scheme proposed by Yang et al., (2008) (i.e., Y08) into the CLM5.0 based on the measured soil properties (EXP2). The results show that the accuracy of soil temperature is slightly improved with an average reduction in RMSE of about 0.1 °C (Table 4). However, there are still large errors during the freeze periods. Therefore, a modified freeze–thaw process by Yang et al. (2018) (i.e., Y18) was incorporated into CLM5.0 (EXP3) based on EXP2. However, the result of the EXP3 experiment was different from what we expected. That is, compared with before improvements, the incorporation of the Y18 scheme increased the simulation error with a larger RMSE value (Table 4).

**Table 4.** The error statistics in simulated soil temperature at different depths at the Tanggula and Beiluhe sites.

| Depth/cm | Experiments | Tanggula | | | Beiluhe | | |
|---|---|---|---|---|---|---|---|
| | | RMSE/°C | R | MBE/°C | RMSE/°C | R | MBE/°C |
| 5 | CTL | 1.91 | 0.99 | 0.40 | 1.35 | 0.99 | 0.72 |
| | EXP1 | 1.76 | 0.99 | 0.12 | 1.41 | 0.99 | 0.75 |
| | EXP2 | 1.64 | 0.99 | −0.14 | 1.30 | 0.99 | 0.60 |
| | EXP3 | 1.72 | 0.99 | −0.36 | 1.35 | 0.99 | 0.59 |
| 10 | CTL | 1.85 | 0.99 | 0.41 | 1.15 | 0.99 | 0.65 |
| | EXP1 | 1.67 | 0.99 | 0.13 | 1.23 | 0.99 | 0.70 |
| | EXP2 | 1.55 | 0.99 | −0.13 | 1.12 | 0.99 | 0.54 |
| | EXP3 | 1.63 | 0.99 | −0.34 | 1.18 | 0.99 | 0.54 |
| 20 | CTL | 1.86 | 0.99 | 0.14 | 1.05 | 0.99 | 0.70 |
| | EXP1 | 1.68 | 0.99 | −0.16 | 1.13 | 0.99 | 0.75 |
| | EXP2 | 1.61 | 0.99 | −0.40 | 1.01 | 0.99 | 0.61 |
| | EXP3 | 1.72 | 0.99 | −0.60 | 1.06 | 0.99 | 0.61 |
| 40 | CTL | 2.01 | 0.99 | −0.08 | 1.15 | 0.99 | 0.86 |
| | EXP1 | 1.83 | 0.99 | −0.37 | 1.24 | 0.99 | 0.90 |
| | EXP2 | 1.79 | 0.99 | −0.59 | 1.06 | 0.99 | 0.76 |
| | EXP3 | 1.91 | 0.99 | −0.79 | 1.14 | 0.99 | 0.76 |
| 80 | CTL | 1.65 | 0.98 | −0.39 | 1.52 | 0.99 | 1.11 |
| | EXP1 | 1.66 | 0.98 | −0.63 | 1.61 | 0.99 | 1.15 |
| | EXP2 | 1.68 | 0.98 | −0.84 | 1.43 | 0.99 | 1.04 |
| | EXP3 | 1.80 | 0.98 | −1.03 | 1.55 | 0.99 | 1.03 |
| 210 | CTL | 2.74 | 0.90 | −1.15 | 1.98 | 0.93 | 1.25 |
| | EXP1 | 2.82 | 0.88 | −1.33 | 2.07 | 0.92 | 1.24 |
| | EXP2 | 2.84 | 0.89 | −1.49 | 1.96 | 0.92 | 1.17 |
| | EXP3 | 3.00 | 0.89 | −1.65 | 2.01 | 0.92 | 1.13 |

Figure 3 illustrates the temporal behavior of soil temperature between simulated and observed at the Beiluhe site and the error metrics are also presented in Table 4. Similar to the Tanggula site, CLM5.0 can also simulate the soil temperature dynamics at the Beiluhe site well and its performance is better at the Beiluhe site. Different from the Tanggula site, however, the soil temperature of the Beiluhe site is slightly overestimated (MBE > 0 °C) during the freezing period. Moreover, we also noticed that the simulation accuracy in soil temperature was worse after replacing the measured soil properties (EXP1) at the Beiluhe site with a greater RMSE value (Table 4). The performance of the thermal roughness length (EXP2) and freeze–thaw process (EXP3) schemes are similar to that of the Tanggula site. That is, the incorporation of the Y08 scheme can improve the simulation results of the soil temperature at the Beiluhe site, while Y18 cannot.

*3.2. Soil Moisture*

Additionally, we also evaluated the simulation performance of CLM5.0 in terms of soil moisture. Figure 4 presents the soil moisture changes between observed and simulated values from 5 to 40 cm at the Tanggula site and the related statistics metrics are listed in Table 5. CLM5.0 can capture the dynamic patterns in soil moisture at the Tanggula site. It is worth noting that the soil moisture in the shallow layer is overestimated (Figure 4a,b), whereas that of the deep layer is underestimated (Figure 4c,d and Table 5) during the thawing period. During the freezing period, the soil moisture is systematically underestimated by the CLM5.0, except for the 5 cm depth.

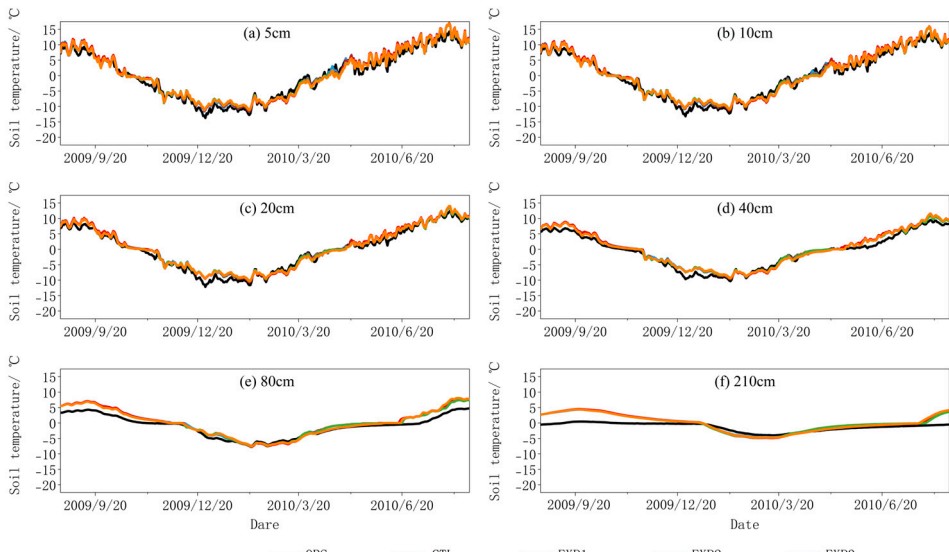

**Figure 3.** Soil temperature changes between observed and simulated performance at different depths at the Beiluhe site.

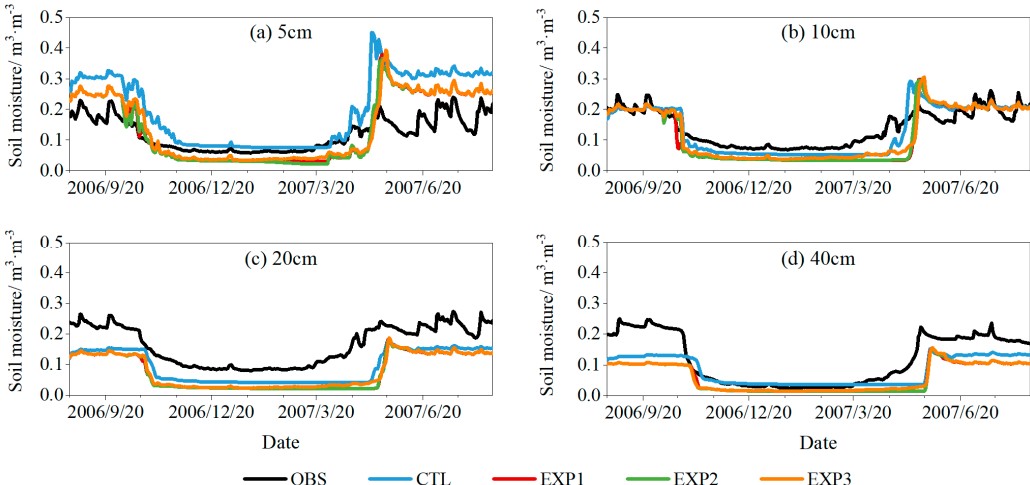

**Figure 4.** Soil moisture changes between observed and simulated performance at different depths at the Tanggula site.

Similarly, we further analyzed the effects of soil properties (EXP1), thermal roughness length (EXP2), and freeze–thaw processes (EXP3) on soil moisture. As shown in Figure 4, the measured soil properties result in smaller soil moisture values than those of the default data, especially at a depth of 5 cm. Therefore, the modified soil properties' data can effectively improve the simulation accuracy of soil moisture at a 5 cm depth, while the errors at other depths are slightly increased (Table 5). Compared with the soil property, the modified thermal roughness length (EXP2) and freeze–thaw process (EXP3) have almost no effect on the simulations in soil moisture, except for the freeze–thaw change phase at a 10 cm soil depth (Figure 4 and Table 5).

The change characteristics in soil moisture at the Beiluhe site are shown in Figure 5. It is observed that the performance of CLM5.0 in soil moisture at the 5 cm depth at the Beiluhe site is similar to that at the Tanggula site. Compared with the Tanggula site, however, soil moisture was significantly underestimated during the thawing period and slightly overestimated during the freezing period at the Beiluhe site. Moreover, soil moisture at the Beiluhe site is insensitive to soil properties (EXP1), thermal roughness length (EXP2), and the freeze–thaw process (EXP3).

**Table 5.** The error statistics in simulated soil moisture at different depths at the Tanggula and the Beiluhe sites.

| Depth/cm | Experiments | Tanggula | | | Beiluhe | | |
|---|---|---|---|---|---|---|---|
| | | RMSE/$m^3 \cdot m^{-3}$ | R | MBE/$m^3 \cdot m^{-3}$ | RMSE/$m^3 \cdot m^{-3}$ | R | MBE/$m^3 \cdot m^{-3}$ |
| 5 | CTL | 0.107 | 0.89 | 0.08 | 0.087 | 0.92 | 0.07 |
| | EXP1 | 0.070 | 0.88 | 0.02 | 0.076 | 0.93 | 0.05 |
| | EXP2 | 0.070 | 0.88 | 0.02 | 0.079 | 0.91 | 0.05 |
| | EXP3 | 0.070 | 0.87 | 0.02 | 0.081 | 0.92 | 0.06 |
| 10 | CTL | 0.036 | 0.91 | −0.01 | 0.069 | 0.92 | −0.01 |
| | EXP1 | 0.051 | 0.89 | −0.03 | 0.077 | 0.92 | −0.04 |
| | EXP2 | 0.050 | 0.89 | −0.02 | 0.076 | 0.92 | −0.04 |
| | EXP3 | 0.044 | 0.90 | −0.02 | 0.076 | 0.92 | −0.04 |
| 20 | CTL | 0.075 | 0.89 | −0.07 | 0.114 | 0.94 | −0.06 |
| | EXP1 | 0.093 | 0.87 | −0.09 | 0.118 | 0.95 | −0.08 |
| | EXP2 | 0.093 | 0.87 | −0.09 | 0.117 | 0.95 | −0.08 |
| | EXP3 | 0.089 | 0.90 | −0.08 | 0.118 | 0.95 | −0.07 |
| 40 | CTL | 0.058 | 0.89 | −0.04 | 0.137 | 0.90 | −0.07 |
| | EXP1 | 0.075 | 0.90 | −0.06 | 0.135 | 0.91 | −0.08 |
| | EXP2 | 0.075 | 0.90 | −0.06 | 0.134 | 0.92 | −0.08 |
| | EXP3 | 0.073 | 0.91 | −0.06 | 0.134 | 0.91 | −0.08 |

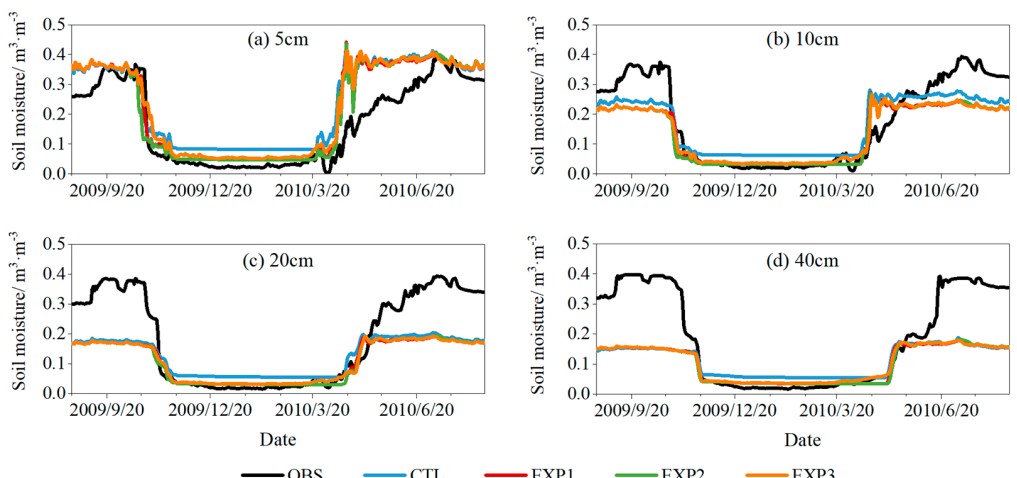

**Figure 5.** Soil moisture changes between observed and simulated performance at different depths at the Beiluhe site.

### 3.3. Soil Thermal Conductivity

Apart from the soil temperature and soil moisture, soil thermal conductivity is also one of the important parameters affecting the hydrothermal condition of permafrost [37]. Due to the lack of measured data [32], we only used soil thermal conductivity at a depth of 5 cm to investigate the simulated performance of soil thermal conductivity by CLM5.0 and three sensitivity experiments. Figure 6 shows the change in soil thermal conductivity at the Tanggula and Beiluhe sites between the observed and simulated performance. Table 6 presents the statistical metrics of the soil thermal conductivity at the two sites.

Soil thermal conductivity during the thawing period can be simulated well by CLM5.0, but it was significantly overestimated during the freezing period. It should be noted that the measured soil thermal conductivity shows opposite characteristics. That is, the soil thermal conductivity during the freezing period is slightly lower than that during the thawing period (Figure 6). The sensitivity experiments on soil thermal conductivity show that soil thermal conductivity during the thawing period is insensitive to soil properties (EXP1), thermal roughness length (EXP2), and the freeze–thaw process (EXP3). However, there is

a large difference in the sensitivity of soil thermal conductivity to the three experiments during the freezing period. Overall, modified soil properties produce larger values of soil thermal conductivity during the freezing period. Moreover, the improved thermal roughness length scheme (EXP2) further contributed to the overestimation of soil thermal conductivity during the freezing period. Based on the EXP2, the modification of the freeze–thaw process (EXP3) can effectively reduce errors in soil thermal conductivity during the freezing period, especially at the Tanggula site with an accuracy of about 23.5% (Table 6). However, the errors in the soil thermal conductivity simulated by the modified CLM5.0 are larger than those using the default parameters. Overall, the simulation error of soil thermal conductivity is larger than that of the soil temperature and moisture, especially during the freezing period.

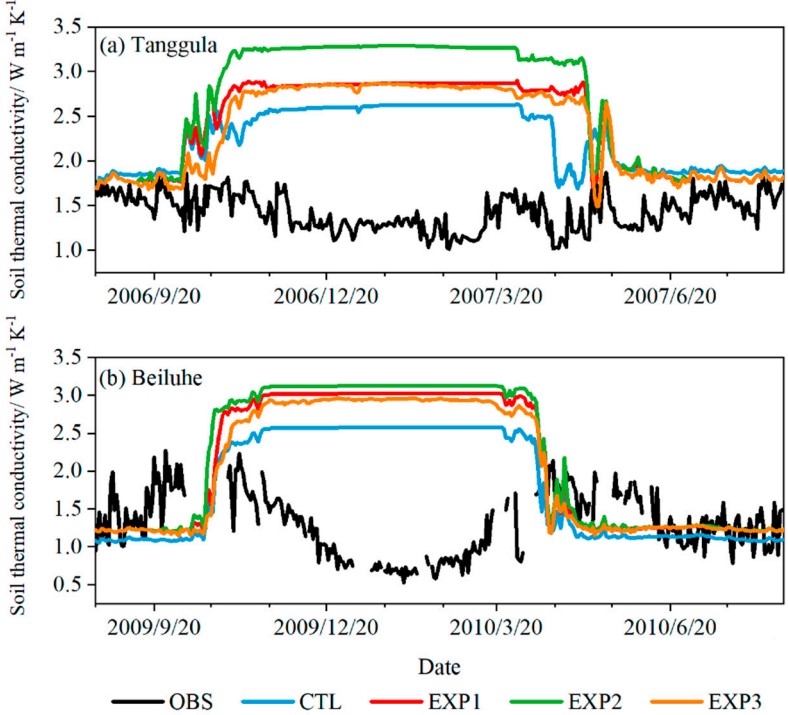

**Figure 6.** Soil thermal conductivity changes between observed and simulated performance at 5 cm depths at the Tanggula and Beiluhe sites.

**Table 6.** The error statistics in simulated soil thermal conductivity at a 5 cm depth at the Tanggula and Beiluhe sites.

| Period | Experiment | Tanggula Site | | | Beiluhe Site | | |
|---|---|---|---|---|---|---|---|
| | | RMSE /W·m$^{-1}$·K$^{-1}$ | R | MBE /W·m$^{-1}$·K$^{-1}$ | RMSE /W·m$^{-1}$·K$^{-1}$ | R | MBE /W·m$^{-1}$·K$^{-1}$ |
| Freezing | CTL | 1.21 | −0.33 | 1.17 | 1.49 | −0.51 | 1.40 |
| | EXP1 | 1.48 | −0.40 | 1.46 | 1.92 | −0.53 | 1.85 |
| | EXP2 | 1.87 | −0.44 | 1.85 | 2.01 | −0.49 | 1.95 |
| | EXP3 | 1.43 | −0.50 | 1.39 | 1.83 | −0.65 | 1.75 |
| Thawing | CTL | 0.49 | 0.05 | 0.43 | 0.45 | 0.28 | −0.32 |
| | EXP1 | 0.62 | −0.44 | 0.46 | 0.37 | 0.28 | −0.19 |
| | EXP2 | 0.70 | −0.46 | 0.50 | 0.37 | 0.29 | −0.18 |
| | EXP3 | 0.56 | −0.42 | 0.40 | 0.39 | 0.04 | −0.20 |
| ALL | CTL | 0.93 | −0.35 | 0.81 | 1.06 | −0.41 | 0.47 |
| | EXP1 | 1.15 | −0.44 | 0.97 | 1.33 | −0.40 | 0.75 |
| | EXP2 | 1.43 | −0.45 | 1.2 | 1.39 | −0.40 | 0.8 |
| | EXP3 | 1.10 | −0.48 | 0.91 | 1.27 | −0.43 | 0.69 |

## 4. Discussion

### 4.1. The Effect of Soil Properties on the Results

Previous studies have shown that soil organic matter significantly impacted soil moisture [34,47]. Organic matter in the soil increases soil porosity, which leads to a high soil moisture content, especially in shallow soil layers where organic matter is concentrated [27,38]. Some studies have been conducted by incorporating organic matter into the soil thermal conductivity scheme in LSMs to improve the accuracy of simulated soil moisture [30,34]. However, soil organic matter also has high hydraulic conductivity, which permits incident precipitation to quickly permeate through the topsoil layers and leads to a dryer surface layer and weaker evaporation [38]. Accounting for the important influence of organic matter on soil hydrothermal parameters, Lawrence et al. (2008) incorporated organic matter into the CLM, including the soil thermal conductivity and soil hydraulic conductivity schemes [27], and this improvement has been retained in CLM5.0.

However, based on the improved model with organic matter (i.e., CLM5.0), there are still large errors between the simulated results and the measured results. For example, we found that soil moisture in the shallow layers was overestimated (Figures 4a and 5a). This is often referred to as "wet bias", and similar results were also indicated by other studies [29]. This may be related to the large difference between the default soil property data and the measured data [47]. In this study, we modified soil properties with measured data to replace default data (EXP1). We found that modified soil properties significantly improved the simulation error of soil moisture in the topsoil layer during the thawing period (Figures 4a and 5a). Moreover, the simulated soil temperature at the Tanggula site was also improved (Table 3). However, the soil moisture errors for other soil depths were larger than those before the modification.

Note that the soil temperature simulation performance at the Beiluhe site was better than that of the Tanggula site, while the moisture simulation has a contrary performance at these two sites (Table 4). This may be related to the local topography. The Tanggula site has a high altitude and is surrounded by mountains and glaciers. Therefore, the spatial heterogeneity is greater than that of the Beiluhe site. Moreover, low vegetation cover and large soil particles at the Tanggula site resulted in less soil moisture and greater susceptibility to external influences. However, these are not well-reflected in the model. In addition, we found that the soil temperature was slightly overestimated during the freezing period at the Beiluhe site (Figure 3), because of the high organic matter content (Table 2). Organic matter improves the water-holding capacity of the soil and increases the water content [34], thus offsetting the decrease in soil temperature during the freezing process, which is somewhat higher compared to low organic soils. It is also related to the underestimation of soil thermal conductivity (Figure 6). Smaller thermal conductivity can result in more heat being stored in the soil, which causes a warmer soil temperature during the period. Furthermore, porosity determines the maximum water stored in the soil and has an important effect on soil hydrothermal parameters [48]. Previous studies have demonstrated that porosity played a more important role in reducing model errors than that of the other soil properties [34,48]. Soil particle size affects the porosity of the soil. Most of the land surface models consider mineral soils and not gravel. Studies indicated that incorporating gravel into the LSMs can significantly reduce the errors in soil temperature and moisture [48]. Previous studies denominated that the porosity of the coral fragment ranged from 0.206 to 0.302, which is less than those of mineral soil in CLM (0.37–0.48) [48]. Gravel content of the Beiluhe site (approximately 60%) is larger than that of the Tanggula site (approximately 24%). However, the gravel is not considered in CLM5.0, which caused significantly underestimated soil moisture in deeper layers at the Beiluhe site, and further influenced soil thermal conductivity and temperature.

### 4.2. The Effect of Snow on the Results

The soil temperature during the freezing period was underestimated at the Tanggula site (Figure 2). Similar simulated errors have been present in most LSMs, and they were

called the "cold bias" of soil temperature [30]. Some studies have indicated that this may be caused by the unsuitable aerodynamic impedance scheme of the model, and it can be effectively reduced by improving the aerodynamic impedance scheme [49]. In addition, snow has an important influence on soil temperature owing to its high albedo, low thermal conductivity, and high latent heat, including snow cover, snow depth, and snow duration [38,50]. Although the effect of snow has been well-incorporated into the CLM, the latest version (CLM5.0) has further improved the snow density and increased the number of snow layers [22,51]. However, these improvements are mainly for the high-latitude regions, where the snow is widespread, long-lasting, and thick [51]. Note that, due to the high wind speed and dry climate on the QTP, the duration of snow is short, and its thickness is thin [52]. Previous studies have indicated that snow cover fractions and the surface albedo were overestimated on the QTP by the CLM4.5 [53], which causes less energy to enter the soil, and the soil temperature is underestimated. Therefore, the "cold bias" of soil temperature can be reduced by optimizing the snow cover scheme. Recently, Li et al., (2020) incorporated Gordon's scheme into Noah–MP and revealed that the simulation of surface albedo was effectively improved, and the "cold bias" of soil temperature was also improved [30]. Apart from the snow scheme, a challenge for snow simulation on the QTP is the effect of wind on snow, which is not accounted for by most current LSMs. Some works have coupled a blowing snow scheme into the CLM4.5, and their results showed that the snow cover, snow depth, and surface albedo can be better reproduced by an improved model [53]. However, the special climatic conditions of the QTP result in the measured snow data being very difficult to obtain, especially in permafrost regions, which limits the study of snow cover and its effects on soil. It should be focused on enhancing technology and networks of snow observations on the QTP in the future.

### 4.3. The Effect of Parameterization Schemes and Other Possible Factors

Generally, the thermal conductivity of ice (2.29 $W \cdot m^{-1} \cdot K^{-1}$) is four times larger than that of water (0.57 $W \cdot m^{-1} \cdot K^{-1}$) [18], which results in a larger soil thermal conductivity during the freezing period than that during the thawing period. However, our results showed the opposite trend (Figure 6), and similar phenomena have been observed in other regions [18,54]. It may be attributed to many small bubbles and spaces within the permafrost, that have a much lower thermal conductivity than water or ice [18]. The initial freezing water content is also a very important factor. Previous studies have indicated that it occurred when the initial moisture content was above 0.195 $m^3 \cdot m^{-3}$ [37], and this conclusion has subsequently been confirmed by Du et al. (2020) [54]. Although the thresholds may be different for different regions, the initial freezing water content certainly has a significant effect on soil thermal conductivity.

Compared with EXP1 (soil properties), EXP2 (Y08, thermal roughness length) and EXP3 (Y18, freeze–thaw process) produced quite distinct soil thermal conductivity during the freezing period (Figure 6), while the simulated soil moisture and soil temperature were similar among the three experiments, which was mainly related to soil moisture. The Y08 scheme affects the thermal parameters by regulating the energy entering the soil, whereas the Y18 scheme affects the ice–water phase change process by changing the freezing temperature and conversion efficiency. Previous studies have demonstrated that slight changes in hydraulic parameters can induce notable changes in thermal parameters [55]. Moreover, soil thermal conductivity is highly sensitive to soil moisture, especially during the freezing period [37], and it has greater uncertainty than soil temperature. Therefore, the difference in soil thermal conductivity between Y08 and Y18 solutions is large.

The soil thermal conductivity scheme in LSMs has a critical impact on the results. Previous studies have indicated that different soil thermal conductivity schemes employed in LSMs can result in a 1–3 °C error in soil temperature [33]. Currently, there are more than 40 soil thermal conductivity schemes that have been proposed and improved for specific study areas, soil types, and soil freeze–thaw conditions [17,56]. Many studies have indicated that soil thermal conductivity was significantly overestimated by the Farouki

scheme on the QTP [32,54]. Based on the Farouki scheme, Lawrence et al., (2008) incorporated organic matter into the scheme and this improved scheme remained in CLM5.0 [27]. However, studies have shown that soil thermal conductivity was still overestimated in CLM5.0 [32] and our results show a similar error (Figure 6). Due to the scarcity of soil thermal conductivity data and the fact that most schemes were not initially proposed for permafrost, most soil thermal conductivity schemes have large uncertainties in permafrost regions of the QTP [57]. Recently, Du et al., (2022) proposed a new dry soil thermal conductivity scheme for permafrost on the QTP based on measurement data [58]. It is not clear whether this scheme will give better results when it is incorporated into the LSMs. In addition, solid thermal conductivity porosity and quartz content of the LSMs still have large errors in permafrost regions of the QTP. The QTP has great spatial heterogeneity, and therefore it is urgent to develop a soil thermal conductivity scheme suitable for the QTP or integrate multiple schemes based on their advantages. Note that soil thermal conductivity is strongly dependent on soil moisture content, so the accuracy of soil moisture content directly determines the accuracy of soil thermal conductivity [17,37]. The soil temperature during the thawing period was overestimated at two sites (Table 3). This may be related to the overestimation of soil thermal conductivity. Soil thermal conductivity determines the rate of heat entering the soil [37]. The greater soil thermal conductivity during the thawing causes heat to enter quickly, and soil temperature in the shallow layer was greater than the other layers, which led to larger temperature gradients. The temperature gradient increases with increasing depth, resulting in more heat entering the deeper layers.

The error in soil moisture was large, although we have modified soil properties. The main reason is the incorporation of a dry surface layer (DSL) soil evaporation resistance in CLM5.0 [59]. Recently, some studies have suggested that the soil evaporation resistance parameterization based on the DSL can lead to an overestimation in soil moisture in the topsoil layer on the QTP [31]. There is no significant linear relationship in soil moisture between the different layers, and influenced by many factors, there are processes such as horizontal runoff and infiltration [60]. Moreover, the hydraulic conductivity scheme in LSMs has a critical impact on soil moisture [61,62]. In addition, unlike other regions, repeated freeze–thaw processes and phase change exist in the active layer in permafrost regions [63]. Hence, an accurate description of the soil freeze–thaw processes in the model is essential for the investigation of the soil hydrothermal regime. Some works have been conducted to improve the simulation accuracy of the freeze–thaw process by modifying freezing points and freezing fronts [26,46]. However, there are still large uncertainties in the results, and it is not clear whether these improvements are generalizable to other regions or models. There is a special phenomenon in permafrost regions in which liquid water still exists at negative temperatures (i.e., unfrozen water) [64]. Although unfrozen water has been accounted for in CLM5.0, our results showed that unfrozen water was underestimated or overestimated at the two sites (Figures 4 and 5). The error in soil moisture during the freezing period can also affect energy and water transport during the thawing period. In recent years, many unfrozen water schemes have been proposed, and their performance was different in different regions [64]. Previous studies have demonstrated that the scheme proposed by Zhang et al., (2017) performs better than others on the QTP [64,65]. A major reason is that this scheme is based on measured data from the QTP [66]. This scheme should be incorporated into the LSMs when conducting a study of the QTP in the future.

It is worth noting that the Y08 scheme can improve the simulation of soil temperature, but it had little effect on soil moisture, and worse, it increased the error in soil thermal conductivity. This is related to the structure of the model. CLM5.0 is a complex and highly coupled system, and modifications of one process have a positive impact on other processes, but it also exposes problems in other parts of the model [22]. In addition, a shallow soil depth and zero flux at the lower boundary in LSMs might lead to unrealistic simulations of soil temperature for long timescales and deep soil layers [67,68], which could affect projections of permafrost and soil carbon stability. Some studies suggested that short-term simulations require a soil depth of at least 40 m [68]. Zero flux at the lower

boundary can be used for shallow-layer simulations since they are mainly influenced by the atmosphere [68,69]. CLM5.0 increased the soil depth to 42 m [22], and in this study, the maximum simulated depth was 210 cm, and the time series was only 1–2 years. Therefore, the influence of soil depth and the lower boundary of the model can be ignored in this study. However, one should be careful when a simulation is conducted with a long timescale or deep soil temperature. Furthermore, there is an error accumulation in the model, and the error will be larger with increasing depth [70]. Thus, the error in deep soil is larger than that in shallow soil (Figure 2). The mismatch between the measured soil depth and the model is also a factor that cannot be ignored.

## 5. Conclusions

In this study, we evaluated the performance of CLM5.0 in simulating soil hydrothermal dynamics using measured data derived from two in situ sites in permafrost regions on the QTP. Moreover, to investigate the influence of soil properties, thermal roughness, and the freeze–thaw process on the simulation results, we also conducted sensitivity experiments. The results showed that CLM5.0 can simulate the characteristics of soil hydrothermal changes at different soil depths in permafrost regions on the QTP. In comparison, the performances in soil temperature and soil thermal conductivity during the thawing period were better than those of soil moisture and soil thermal conductivity during the freezing period. However, soil temperature during the freezing period and soil moisture during the thawing period were significantly underestimated, and the error increased with increasing depth. Soil thermal conductivity was remarkably overestimated. Modified soil properties and the thermal roughness length (Y08) scheme can slightly improve the simulation accuracy of soil temperature and soil moisture, and the freeze–thaw process (Y18) has a positive impact on the soil thermal conductivity, especially the freezing period. It is worth noting that there are still large errors in soil moisture and soil thermal conductivity, although some improvements have been conducted in this study. The main reason is that the key hydrothermal parameterizations in the model have large uncertainties in study sites. In the future, it is necessary to develop soil thermal conductivity, hydraulic conductivity, and snow schemes by combining a large number of observations on the QTP and coupled into the LSMs.

**Author Contributions:** Conceptualization: R.L. and S.Y.; formal analysis and investigation: T.W., L.Z., Y.Z., J.S. and Y.Q.; Writing–original draft: S.Y., funding acquisition: R.L. and T.W.; writing—review and editing: S.Y., R.L., L.Z., T.W., X.W. and Y.Z. All authors have read and agreed to the published version of the manuscript.

**Funding:** This work was supported by the National Natural Science Foundation of China (42071093), the National Key Research and Development Program of China (2020YFA0608500), and the foundation of the State Key Laboratory of Cryospheric Science (SKLCS-ZZ-2022).

**Data Availability Statement:** The measured data at the Tanggula and Beiluhe stations are derived from Cryosphere Research Station on the Qinghai-Tibet Plateau, Chinese Academy of Sciences (http://new.crs.ac.cn/). The CLM5.0 model is available from Community Earth System Model (https://www.cesm.ucar.edu/).

**Conflicts of Interest:** The authors declare no conflict of interest.

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
