# Peer review of "Evaluation of the Performance of CLM5.0 in Soil Hydrothermal Dynamics in Permafrost Regions on the Qinghai–Tibet Plateau"

_remotesensing, doi:10.3390/rs14246228_

Round 1
Reviewer 1 Report (Previous Reviewer 2)
The authors have addressed my concerns. I have no further comments.
Author Response
Thanks for your positive comments on our revised manuscript. Again, we appreciate to the reviewer’s constructive and insightful comments and thoughtful suggestions on our original manuscript. These comments and suggestions are very helpful for the improvement of our manuscript.

Reviewer 2 Report (New Reviewer)
The authors have responded well to the previous comments and indeed improved the manuscript. Here I have a few suggestions before the final acceptance:
1. The authors describe the parameterization and model set-up in section 2.2.2, and also discuss their potential influence on the simulation results, but the impacts of boundary condition are not included. Zero flux at the lower boundary (line 209) is an idealized situation, so the setup will affect the simulation accuracy, especially when the lower boundary is not so deep. I suggest the author add a few sentences to discuss the influence.
2. Line 42: Are there some words missing after the “ground temperature”?
3. Line 81: there are two “hydrology” in this sentence
4. Line 92-93: the underestimation in the cold season and overestimation in the warm season only occur in the QTP or they can be found in other regions?
5. Figure 5” a “/” is missing in the last column.
6. Line 348, delete “however”
Author Response
The authors have responded well to the previous comments and indeed improved the manuscript. Here I have a few suggestions before the final acceptance:
- The authors describe the parameterization and model set-up in section 2.2.2, and also discuss their potential influence on the simulation results, but the impacts of the boundary condition are not included. Zero flux at the lower boundary (line 209) is an idealized situation, so the setup will affect the simulation accuracy, especially when the lower boundary is not so deep. I suggest the author add a few sentences to discuss the influence.
Response:
Thanks for the reviewer’s suggestion. We have added some discussions of the low boundary in Line 539-549 in the revised manuscript.
“In addition, a shallow soil depth and zero flux at the lower boundary in LSMs might lead to unrealistic simulations of soil temperature for long timescales and deep soil layers (Alexeev et al., 2007; Hermoso et al., 2020), which could affect projections of permafrost and soil carbon stability. Some studies suggested that short-term simulations require a soil depth of at least 40m (Hermoso et al., 2020). Zero flux at the lower boundary can be used for shallow layer simulations since they are mainly influenced by the atmosphere [Hermoso et al., 2020; Sun et al., 2020]. CLM5.0 increased the soil depth to 42m (Lawrence et al., 2019), and in this study, the maximum simulated depth was 210cm, and the time series s only 1-2 years. Therefore, the influence of soil depth and lower boundary of the model can be ignored in this study. However, it should be careful when a simulation is conducted with a long timescale or deep soil temperature.”
References:
Alexeev, V.A., et al., An evaluation of deep soil configurations in the CLM3 for improved representation of permafrost. Geophysical Research Letters, 2007. 34(9).
Hermoso de Mendoza, I., et al., Lower boundary conditions in land surface models – effects on the permafrost and the carbon pools: a case study with CLM4.5. Geoscientific Model Development, 2020. 13(3): p. 1663-1683.
Sun Z., et al., Influence of lower boundary conditions on the numerical simulation of permafrost temperature field change. Journal of Glaciology and Geocryology, 2021. 43(2): p. 357-369 (In Chinese with English abstract).
- Line 42: Are there some words missing after the “ground temperature”?
Response:
Thanks for the reviewer’s suggestion. The ground temperature here referred to the ground temperature of permafrost. Therefore, we added “of permafrost” after the “ground temperature” in Line 41 in the revised manuscript.
- Line 81: there are two “hydrology” in this sentence.
Response:
Thanks for the reviewer’s suggestion. We have deleted the word “hydrology” in Line 79 in the revised manuscript.
- Line 92-93: the underestimation in the cold season and overestimation in the warm season only occur in the QTP or they can be found in other regions?
Response:
Thanks for the reviewer’s comment and suggestion. Many studies have shown that soil temperature in the cold season was underestimated on the QTP (Li et al., 2020; Yang et al., 2021). Similar errors have been found in other study areas, such as Eurasia (Shukla et al., 2019), India (Bhattacharya and Mandal, 2015), and America (Xia et al., 2013). However, the overestimation of soil temperature in the warm season was rarely found in other regions. Previous studies have also indicated that ERA5-Land soil temperature has a warm bias at high latitudes and a cold bias in mid–low-latitude, high-elevation areas (Cao et al., 2020).
Therefore, we have revised the original sentence to “Moreover, the soil temperature in the cold season was generally underestimated in mid-low-latitude and high-elevation regions, especially the QTP” in Line 90-91 in the revised manuscript.
References:
Bhattacharya, A. and M. Mandal, Evaluation of Noah land-surface models in predicting soil temperature and moisture at two tropical sites in India. Meteorological Applications, 2015. 22(3): p. 505-512.
Cao, B., et al., The ERA5-Land soil temperature bias in permafrost regions. The Cryosphere, 2020. 14(8): p. 2581-2595.
Li, X., et al., Improving the Noah‐MP Model for Simulating Hydrothermal Regime of the Active Layer in the Permafrost Regions of the Qinghai‐Tibet Plateau. Journal of Geophysical Research: Atmospheres, 2020. 125(16).
Shukla, R.P., et al., The Influence of Summer Deep Soil Temperature on Early Winter Snow Conditions in Eurasia in the NCEP CFSv2 Simulation. Journal of Geophysical Research: Atmospheres, 2019. 124(16): p. 9062-9077.
Xia, Y., et al., Validation of Noah-Simulated Soil Temperature in the North American Land Data Assimilation System Phase 2. Journal of Applied Meteorology and Climatology, 2013. 52(2): p. 455-471.
Yang, S., et al., Evaluation of reanalysis soil temperature and soil moisture products in permafrost regions on the Qinghai-Tibetan Plateau. Geoderma, 2020. 377.
- Table 5” a “/” is missing in the last column.
Response:
Thanks for the reviewer’s suggestion. We have added “/” in Table 5 in the last column of the revised manuscript.
- Line 358, delete “however”.
Response:
Thanks for the reviewer’s suggestion. We have deleted “however” in Line 359 in the revised manuscript.

This manuscript is a resubmission of an earlier submission. The following is a list of the peer review reports and author responses from that submission.
Round 1
Reviewer 1 Report
The authors simulated the soil temperature and moisture at two sites using the latest version of CLM model to evaluate its performance in the Qinghai-Tibet plateau, and set 4 experimental runs to explore the factors affecting the hydrothermal simulation results. Overall, the manuscript still needs some editing, and a major revision may be needed. My major concerns are:
1. The current results in this manuscript didn’t provide much new information and additional discussion is needed to highlight the message from this study. The authors only simply describe the simulation results and list some well-known factors related to the model inaccuracy, while the specific discussion is lacking. For example, why the soil temperature simulation results at Beiluhe site is better than that at Tanggula site, while the moisture simulation has a contrast performance at these two sites? I suggest the authors provide a detailed discussion about the simulation results and the model uncertainties, so as to provide some future perspectives on permafrost simulation research in the Tibetan Plateau.
2. The introduction may need re-organization. Currently, the first paragraph does not seem to be well related to the rest of this section. I also suggest the authors conduct a short review about the hydrothermal studies in the QTP and highlight the objective and implications of this study.
3. The author describes two thermal roughness length schemes of CLM5.0 in detail, but some basic information about model setup is still lacking. For example, what is the boundary condition? Is soil organic content considered in the current simulation as formula 7&8 described? If it is, where are the soil organic data from? Does soil texture change along the soil depth in the current simulation?
4. The in-situ data used in the study also need a detailed description. For example, in line 186 and Table 2, the authors emphasize the importance of soil texture in permafrost simulation and replaced the default soil texture with observations in experiments 1-3, but there is no information about soil texture observations at the two study sites.
5. The authors introduced the thermal roughness length schemes in the experimental run. I would like to suggest the authors to compare the temperature at the ground surface layer (0 cm).
6. The comparison of “soil thermal conductivity” is interesting, particularly, that the conductivity decreases during the frozen season. However, I would like to see more details on the derivation of in-situ soil thermal conductivity, and the potential uncertainties. And any data in the deeper soils? I would expect the soil thermal conductivity in the deep layer increases with soil frozen since it should be more saturated than surface soils.
More specifically, here are some minor suggestions:
1. Title: there seems missing a word “Dynamics” after “Soil Thermal”.
2. Line 42-43: an “elevation” is missing
3. Line 49-51: rewrite the sentence after “such as”.
4. Line 86: what does the “highest” mean? Among all the observation sites or just these two?
5. Line 217-224: It is well know known that snow cover affects the soil temperature simulation. Is the snow depth also simulated in this study? If it is, how is the performance and what is the possible influence on the temperature simulation at these two sites?
6. Line 235: “the mismatch between the measured soil depth and the model”. How is the soil stratified in the model and measurements? Are the stratification for in-situ soil temperature, moisture and soil texture the same?
7. Line 242 and line 256: the Y18 scheme is not suitable for both Tanggula and Beiluhe. Is there any discussion on the possible reason?
8. Figure 3: the temperature overestimation during the summer is stronger but not addressed or discussed in the text. The line color scheme in panel e is different from the others.
9. Line 266-267: is the effects of soil organic content considered in soil hydraulic parameter?
10. Figure 6: what is the indicator for different depth? It seems there is only one soil layer in this figure. The observed thermal conductivity is only for surface soil, correct?
11. Table 6: the R at Tanggula site is overall negative while it’s all positive at Beiluhe site? There are two RMSEs at Beiluhe site.
Reviewer 2 Report
This manuscript evaluates the performance of the CLM5.0 land surface model in simulating soil hydrothermal dynamics at two permafrost sites on the Qinghai-Tibet Plateau (QTP). The authors further investigate the effects of alternative parameterizations regarding soil texture, thermal roughness length and freeze-thaw processes on the model results via sensitivity experiments. Since this is a model evaluation paper, I would expect some more insights about the underlying causes of model behaviors and biases. However, the current analysis and discussion are a bit too weak, without providing much new, in-depth information about how to reduce model bias and to improve model performance.
First, there are only two sites and two years’ observation to compare with the model. For the two sites, the model underestimates soil temperature at one site and overestimates soil temperature at the other one. Thus the conclusion that the model has a “cold bias” does not stand. Given that there are now more publicly available measurements of soil temperature and moisture over QTP, I would suggest including more site-years to provide a more general picture of the model performance for QTP.
Second, the explanations for the model biases are too speculative, without further examinations of related variables. For example, you propose that the “simulated error in the snow” might contribute to the cold bias at Tanggula site, but have you checked for the simulated snow cover and/or depth to see whether they are overestimated?
Third, about the soil thermal conductivity (lambda), it is surprising to see that the Y08 thermal roughness length scheme and the Y18 freeze-thaw scheme lead to quite distinct lambda results compared to EXP1, given that the simulated soil moistures (and soil temperatures) are all so close among EXP1~3. Please provide an explanation for the mechanisms through which Y08 and Y18 schemes can affect lambda.
Finally, the current manuscript is not carefully written, with many grammatical errors and typos. An extensive editing of English language is required.
Specific comments:
Abstract Line 29: “… the internal structure and physical processes of the model, such as soil thermal conductivity, hydraulic conductivity, …” – these are parameters, not structure or process.
Lines 151-155: Do you mean, you loop the “first-year data” to do the spin-up? Which year? Besides, there is only one year’s model results (and observations) that are shown in this study, then, why do you emphasize “the last one-year results for analysis”? If you have run a longer simulation (i.e. including some years before the year that is analyzed), what is the climate forcing?
Equation 14: where is the epsilon used?
Line 225-226: Do you use the meteorological measurements at the sites as the climate forcing of CLM, rather than using a global reanalysis product? If so, the forcing cannot explain the biases in the simulated soil hydrothermal variables.
Line 242-243: You cannot conclude that the Y18 scheme is not suitable here merely because the simulated soil temperature is more different from observations, because there can be error compensations in the model.
Line 267-270: If the soil moisture at deeper layer is “significantly underestimated”, why do you call it a “wet bias”? Besides, I don’t find any “wet bias” reported in the cited reference 31.
Line 272 “The soil moisture content at different depths is not related” – this statement is not correct.
Figure 6 caption: “at different depths” should be “at 5 cm depth”?